

# Marine catfishes (Ariidae—Siluriformes) from the Coastal Amazon: mitochondrial DNA barcode for a recent diversification group?

Ítalo Lutz[1], Thais Martins[1], Paula Santana[1], Charles Ferreira[1], Josy Miranda[1], Suane Matos[1], Valdemiro Muhala[2,3], Iracilda Sampaio[3], Marcelo Vallinoto[3] and Grazielle Evangelista-Gomes[1]

[1] Laboratório de Genética Aplicada (LAGA), Instituto de Estudos Costeiros (IECOS), Universidade Federal do Pará, Bragança, Brazil
[2] Divisão de Agricultura, Instituto Superior Politécnico de Gaza, Chókwè, Mozambique
[3] Laboratório de Evolução (LEVO), Instituto de Estudos Costeiros (IECOS), Universidade Federal do Pará, Bragança, Brazil

Corresponding author
Grazielle Evangelista-Gomes,
grazielle@ufpa.br

## ABSTRACT

**Background:** Ariidae species play a significant role as fishing resources in the Amazon region. However, the family's systematic classification is notably challenging, particularly regarding species delimitation within certain genera. This difficulty arises from pronounced morphological similarities among species, posing obstacles to accurate species recognition.
**Methods:** Following morphological identification, mitochondrial markers (COI and Cytb) were employed to assess the diversity of Ariidae species in the Amazon.
**Results:** Our sampling efforts yielded 12 species, representing 92% of the coastal Amazon region's diversity. Morphological identification findings were largely corroborated by molecular data, particularly for species within the *Sciades* and *Bagre* genera. Nonetheless, despite morphological support, *Cathorops agassizii* and *Cathorops spixii* displayed minimal genetic divergence (0.010). Similarly, *Notarius quadriscutis* and *Notarius phrygiatus* formed a single clade with no genetic divergence, indicating mitochondrial introgression. For the majority of taxa examined, both COI and Cytb demonstrated efficacy as DNA barcodes, with Cytb exhibiting greater polymorphism and resolution. Consequently, the molecular tools utilized proved highly effective for species discrimination and identification.

## INTRODUCTION

The order Siluriformes is quite diverse and widely distributed throughout the world, having two families with marine representatives, Ariidae and Plotosidae (*Sullivan, Lundberg & Hardman, 2006*; *Nelson, Grande & Wilson, 2016*). Ariidae is a monophyletic group, represented by 157 valid species, distributed in three subfamilies, Ariinae, Bagreinae

and Galeichthyinae (*Betancur-R, 2009*; *Marceniuk, Menezes & Britto, 2012a*; *Fricke et al., 2023*).

The species are found in tropical and subtropical waters of the Atlantic, Indian and Pacific Oceans (*Kailola, 2004*; *Barletta & Blaber, 2007*). The vast majority of the species are confined to coastal, estuarine and mangrove areas with turbid waters, generally found at a depth of up to 150 m (*Kailola, 2004*; *Marceniuk & Menezes, 2007*).

Ariids have economic value in many regions where they are found (*Amin et al., 2016*; *Marceniuk et al., 2022*). In Brazil, they are common targets for small and large-scale fishing, using different fishing methods (*Dantas et al., 2010*; *Bentes et al., 2012*; *Carvalho Neta et al., 2014*). More precisely on the northern coast of Brazil, species such as *Sciades parkeri*, *Sciades proops* and *Bagre bagre* are important fishing resources (*Martins et al., 2021b*). These species are generally sold in the form of fillets (*Betancur et al., 2015*; *Gilio-Dias et al., 2020*) and supply the national and international market, while the swim bladder is exported, mainly to Asian countries for cosmetic purposes (*Jimenez et al., 2021*). Other species of catfish are also sold fresh in local markets and fairs such as the Feira Livre de Bragança, in the Salgado region and northeast of Pará (*Martins et al., 2021b*).

Commercial fraud resulting from the replacement of "Gurijuba" *S. parkeri* fillets with "Uritinga" *S. proops* fillets has been reported in the coastal regions of the Amazon (*Gomes et al., 2019*). *S. parkeri* is currently listed as vulnerable by the International Union for Conservation of Nature (IUCN). To prevent population decline and ensure environmental and economic sustainability, it is essential to enhance knowledge of the diversity and systematics of catfish in the region and implement effective management and conservation measures for these resources (*Marceniuk et al., 2021*).

Despite the great relevance in the fishing and bioecological scenarios, its systematics is still not well understood. Many studies have been done on the taxonomy of this group, but some genera have unclear species boundaries, which leads to confusion in taxon identification (*Ferraris, 2007*; *Mazlan et al., 2008*; *Betancur-R, Marceniuk & Béarez, 2008*; *Ng, 2012*; *Marceniuk, Oliveira & Ferraris, 2023*). In some cases, Arrids species present marked morphological similarities in addition to ontogenetic variations and sexual dimorphism.

The lack of testimonial specimens in museums and collections, combined with other factors, makes it difficult to identify diagnostic characters and recognize species. Consequently, it is challenging to reconstruct their evolutionary kinship relationships (*Marceniuk, 2005*, *2007*; *Marceniuk & Menezes, 2007*; *Betancur-R, Marceniuk & Béarez, 2008*). However, more recent approaches, utilizing integrative data from morphology and evolutionary molecular genetics, have significantly improved the understanding of the real diversity of Ariidae and their phylogenetic relationships (*Marceniuk, Menezes & Britto, 2012a*; *Marceniuk, Oliveira & Ferraris, 2023*). Markers based on DNA have been efficiently and robustly utilized to identify fish diversity (*de Sousa et al., 2022*; *Tang et al., 2023*; *Lutz et al., 2023a*). They have been instrumental in tasks such as registration of cryptic species (*González-Castellano et al., 2020*; *Louro et al., 2021*), tracing threatened species in trade (*Martins et al., 2021a*), resolving phylogenetic and taxonomic inconsistencies (*Ronco et al., 2020*; *Dornburg & Near, 2021*; *van Staden et al., 2023*), and identifying and authentication

processed products derived from fish (*Veneza et al., 2018*; *Gomes et al., 2019*; *Barbosa et al., 2020*; *Barbosa, Sampaio & Santos, 2021*; *Lutz et al., 2023b*).

Mitochondrial markers, such as the Cytochrome C Oxidase Subunit 1 (COI) and Cytochrome b (Cytb) genes, are frequently employed to identify species based on their genetic differences. These markers can distinguish between closely related taxa with high accuracy, even with single-locus analyses (*Palacios-Barreto et al., 2017*; *Dalton et al., 2020*; *Lutz et al., 2023b*). However, many marine catfish species from the Ariidae family lack molecular data in public databases, which hampers taxonomic and systematic studies of these species. Therefore, our objective was to assess the diversity of Ariidae on the Amazon coast using mitochondrial DNA Barcodes (regions of the COI and Cytb genes) and to provide new insights into the evolutionary relationships of the group.

## MATERIALS AND METHODS

### Sampling

Whole individuals of Ariidae were collected at landing ports and fish markets in Bragança, northeast of the State of Pará, Amazon coast (Lat 1°3′57.43″S, Long 46°47′22.24″W), between January and December 2020. The samples used in this study were obtained directly from fishing docks and fishmongers, thus comprising dead specimens. Therefore, no collection licenses or approval by the by Animal Ethics Committee were required. All samples were collected and transported with the authorization of the Instituto Chico Mendes de Conservação da Biodiversidade (ICMBio) (SISBIO license n. 12773-1).

Entire individuals of Ariidae, including specimens of all collected species, were photographed to form the scientific image bank of the Ariidae family, and underwent the process of fixation with 10% formaldehyde and preservation with absolute commercial alcohol (*Martins, 1994*). After the fixation/preservation process, the samples were incorporated into the zoological collection of the Laboratório de Genética Aplicada (LAGA), Instituto de Estudos Costeiros (IECOS), Universidade Federal do Pará (UFPA).

Muscle tissue samples were collected from each individual. Each tissue was given a number and stored in 2.0 mL Eppendorf microtubes containing 70% alcohol. Soon after, these samples were stored in a freezer at −20 °C at the Laboratório de Genética Aplicada (LAGA), Universidade Federal do Pará (UFPA).

### Morphological analysis

All samples were morphologically identified through literature (*Marceniuk, 2005*, *2007*; *Marceniuk et al., 2017a*). According to *Marceniuk, Oliveira & Ferraris (2023)*, the genera *Aspistor*, *Amphiarius*, and *Notarius* are synonyms, so we identified all species of the three genera as *Notarius*, as validated by the IUCN and Eschmeyer's Catalog of Fishes (*Fricke et al., 2023*).

The morphological analysis strictly followed *Marceniuk (2007)* and *Marceniuk et al. (2017a)*, with updates to genus and species names. The individuals were observed from the left side, with measurements taken using a manual caliper (0.1 mm) and rulers adjusted to millimeters. Among the main characters observed, we used: shape and length of the maxillary barbel, number of pairs of mental barbels, number of rays in the anal fin, size and

length of the base of the adipose fin, size of the eye in relation to the body, thin or thick lips, shape of the nuchal plate, presence of the dental plate in the vomer, shape of the accessory dental plates and number of rays in the first or second gill arch. The characteristics are described in detail in the literature consulted (*Marceniuk, 2007*; *Marceniuk et al., 2017a*).

## Laboratory procedures

We followed the saline solution (NaCl) protocol by *Aljanabi & Martinez (1997)* to extract DNA from the samples. We then added 2 µL of a solution with Gel red™ dye and Blue Juice to the extraction products, and ran them on a 1% agarose gel with electrophoresis at 60V for about 40 min, to check the quality of the DNA.

We amplified the Barcode region of the COI gene and the first part of the Cytb gene from the 5′ end, from the total genomic DNA, using the Polymerase Chain Reaction (PCR) technique. We used the primers FishR1 and FishR2 (*Ward et al., 2005*) for COI and FishCytb-F and TrucCytb-R (*Sevilla et al., 2007*) for Cytb. We followed the same PCR volumes and conditions as *Gomes et al. (2019)* for *S. parkeri* and *S. proops*.

We purified the positive PCR products with PEG 8000 (Polyethylene Glycol), as described by *Paithankar & Prasad (1991)*, and sequenced them using the dideoxyterminal method (*Sanger, Nicklen & Coulson, 1977*), with the Big Dye 3.1 Kit (ABI Prism TM Dye Terminator Cycle Sequencing Ready Reaction-PE; Thermo Fisher, Waltham, MA, USA), according to the manufacturer's instructions. We then performed capillary electrophoresis ABI 3500 XL automatic sequencer (Thermo Fisher, Waltham, MA, USA).

## Genetic sequence and analysis bank

To investigate the genetic diversity and phylogenetic relationships of Ariidae, we constructed a genetic sequence and analysis bank using two mitochondrial genes, COI and Cytb. We collected 57 specimens from 12 species and four genera of Ariidae, and identified them morphologically. We sequenced both genes for each specimen and obtained 114 sequences in total. The sequences generated were submitted to GenBank. The COI sequences have accession numbers OR646749 to OR646805 and the Cytb sequences have accession numbers OR674791 to OR674847. In Barcode of Life Data System (BOLD), the COI sequences have been deposited under accession numbers BAGR001-24 to BAGR057-24. Six sequences were added to this database to compose the outgroup (Ictaluridae *Ictalurus punctatus*-NC003489, Auchenipteridae *Tympanopleura atronasus*—MT712806, Pimelodidae *Brachyplatystoma rousseauxii*—MT551779—COI; *I. punctatus*-AY184253, *T. atronasus*-DQ119403, *B. rousseauxii*-JF898518-Cytb), available on GenBank. The generated sequences were automatically aligned in the ClustalW application (*Thompson, Higgins & Gibson, 1994*) and corrected in BioEdit 7.2.5 (*Hall, 1999*). Aligned sequences were concatenated using Sequence Matrix 1.9 (*Vilgalys & Hester, 1990*). Possible stop codons were investigated using the MEGA 11.0.11 (*Tamura, Stecher & Kumar, 2021*).

Interspecific and intraspecific Kimura-2-parameter (K2P) distances for Ariidae were calculated in MEGA 11.0.11 (*Tamura, Stecher & Kumar, 2021*) for the two databases, COI and Cytb. The maximum and minimum K2P distances were visualized graphically using histograms generated in the R 4.3.0 program (*R Core Team, 2023*), with the ggplot2

package (*Wickham, Chang & Wickham, 2022*). The barcoding gap for each species was calculated as the difference between the minimum interspecific distance and the maximum intraspecific distance (*Cardoni et al., 2015*; *Wei et al., 2021*).

The two databases were analyzed using the DNAsp (*Librado & Rozas, 2009*) to identify the frequency and possibility of sharing haplotypes between valid species, previously identified with morphology.

Using GenBank (National Center for Biotechnology Information—http://www.ncbi.nlm.nih.gov) (*Benson et al., 2012*) and the BOLD platform (Barcoding of Life Database—https://www.boldsystems.org/) (*Ratnasingham & Hebert, 2007*) the genetic similarity of the haplotypes identified with the available sequences was verified.

The choice of the best evolutionary model was conducted on the CIPRES Science Gateway 3.3 platform (*Miller, Pfeiffer & Schwartz, 2010*) for both data sets, using jModelTest2 in XSEDE (*Darriba et al., 2012*; *Towns et al., 2014*). The analysis recommended the HKY+I+G evolutionary model for the two markers (COI and Cytb), based on the Bayesian Information Criterion (BIC). The Bayesian inference tree with the concatenated bank was built in the MrBayes 3.2 software in XSEDE 3.2.7 (*Ronquist et al., 2012*), using 50 million generations and 10% burn-in, being conducted on the CIPRES Science Gateway 3.3 platform (*Miller, Pfeiffer & Schwartz, 2010*). The log files were checked in Tracer 1.5 (*Rambaut & Drummond, 2009*) and the convergence chains considered adequate presented values greater than >200 ESS (Effective Sampling Size).

We applied the software BEAST 1.8.4 (*Drummond & Rambaut, 2007*; *Rambaut & Drummond, 2009*) to perform Bayesian inference (BI) analyses on the COI and Cytb datasets separately. We ran the analyses on the CIPRES Science Gateway 3.3 platform (*Miller, Pfeiffer & Schwartz, 2010*), using a strict clock model and the Yule speciation process as the tree prior. The posterior probability was estimated with 50 and 30 million generations (COI and Cytb, respectively) and 10% burn-in. The log files were checked in Tracer 1.5 (*Rambaut & Drummond, 2009*) to evaluate the convergence chain and the appropriate burn-in length. The convergence chains considered adequate presented values greater than >200 ESS. For the analysis of the genus *Notarius*, public sequences available on GenBank for *Notarius phrygiatus* and *Notarius quadriscutis* were added to the COI bank (Cytb sequences were not available). A Bayesian inference analysis was performed on this bank, repeating almost all of the previous steps, only changing the recommended model to HKY +I. The trees generated in BEAST were summarized in TreeAnnotator 1.10.4 to obtain the best tree. The resulting tree was visualized in FigTree 1.4.4 (http://tree.bio.ed.ac.uk/software/figtree/) and edited in InkScape 0.92.4 (https://www.inkscape.org).

## Species delimitation

For species delimitation tests, three methods were applied: a) Generalized Mixed Yule Coalescent, GMYC (*Pons et al., 2006*), b) Bayesian Poisson Tree Process, PTP (*Zhang et al., 2013*) and c) Assemble Species by Automatic Partitioning, ASAP (*Puillandre, Brouillet & Achaz, 2021*). The designation Molecular Operational Taxonomic Units (MOTUs) (*Floyd et al., 2002*) was taken into consideration as molecular units to represent potential species.

The GMYC analysis was conducted using the Splits package (*Ezard, Fujisawa & Barraclough, 2009*) in the R 4.3.0 program (*R Core Team, 2023*). The analysis was performed with a summarized ultrametric tree from the BEAST program.

The PTP method was conducted using as input file a maximum likelihood phylogenetic tree in the IQ-TREE 2.1.3 software (*Nguyen et al., 2015*), with the GTR F model with a total of 1,000 Boostrap pseudoreplicates (*Felsenstein, 1985*). The analysis was carried out on the PTP online platform (https://species.h-its.org/ptp/), following the platform's standard parameters.

ASAP analysis was performed on the ASAP online platform (https://bioinfo.mnhn.fr/abi/public/asap/asapweb.html) with the Kimura (K80) TS/TV evolutionary model.

# RESULTS

## Species diversity, based on morphology

For all 57 entire individuals collected, we performed identification based on morphology, consulting specialized literature on the group. We recorded a total of 12 species, five species of the *Sciades* genus: *S. couma* (N = 6), *S. herzbergii* (N = 5), *S. parkeri* (N = 6), *S. passany* (N = 4) and *S. proops* (N = 4); four from the genus *Notarius*: *N. grandicassis* (N = 5), *N. phrygiatus* (N = 3), *N. quadriscutis* (N = 5) and *N. rugispinis* (N = 5); two from the genus *Cathorops*: *C. agassizii* (N = 4) and *C. spixii* (N = 5); and one from the genus *Bagre*: *B. bagre* (N = 5).

## Species diversity, based on DNA markers and comparison with public banks

We analyzed the diversity of Ariidae catfishes from the Amazon coast using 114 sequences of two molecular markers (COI and Cytb genes). Each marker had 57 sequences from different specimens. The sequences were free of insertions, deletions or stop codons, indicating that they were functional fragments for both markers. The COI sequences had a length of 505 bp, corresponding to the Barcode region, while the Cytb sequences had a length of 704 bp, resulting in a total length of 1,209 bp for the concatenated dataset.

The COI and Cytb markers revealed 25 and 31 haplotypes, respectively. Most species had unique haplotypes, such as *C. agassizii* and *C. spixii*. However, *N. phrygiatus* and *N. quadriscutis* shared haplotypes. These two species had varying numbers of individuals with common haplotypes depending on the marker, with five individuals for COI and two individuals for Cytb, indicating a single species (Table S1).

For molecular identification based on genetic similarity, we observed that some comparisons between COI haplotypes and sequences from public databases showed inconsistencies with the previous identification based on morphology (Table S2). The haplotypes of *N. phrygiatus* and *N. quadriscutis* obtained a similarity of 100% and 99.8% with *Notarius luniscutis* in GenBank, respectively. In BOLD, the haplotypes of *N. phrygiatus* and *N. quadriscutis* obtained a similarity of 100% and 99.79% with *N. grandicassis*, respectively. The haplotypes of *C. agassizii* were identified as *C. spixii* in the two public database, *C. agassizii* 4 and *C. agassizii* 7 showed similarities of 99.37% and 100% in GenBank, respectively. In BOLD, *C. agassizii* 4 obtained 99% and *C. agassizii*

798.8%. The *S. parkeri* 3 haplotype was identified as *Netuma* sp. in GenBank, however, it obtained 100% similarity in BOLD with its respective morphological identification, *S. parkeri*. The *S. couma* and *S. passany* haplotypes that did not match in GenBank were identified with more than 100% similarity in BOLD with their respective morphological identifications (Table S2).

Cytb marker haplotypes also showed inconsistencies in comparisons made in GenBank (Table S3). The haplotypes of *N. phrygiatus* and *N. quadriscutis* returned with similarity greater than 99% for the species *N. quadriscutis* and *N. luniscutis*, respectively. The *C. agassizii* haplotypes showed a similarity greater than 99% with *C. spixii* and *Cathorops arenatus* species. All *C. spixii* haplotypes returned with a similarity greater than 99% to the species of *C. spixii* and *C. arenatus*. The haplotypes of the species of the genera *Bagre* and *Sciades*, and *N. grandicassis* specie were correlated with more than 99% with their respective morphological identifications (Table S3).

Using the combined COI and Cytb database, the BI tree identified 10 clades with strong statistical support, especially for the *Sciades* genus and the *B. bagre*, *N. rugispinis* and *N. grandicassis* species (Fig. 1). The *A. phrygiatus* and *As. quadriscutis* species formed a single clade with high support, as well as *C. agassizii* and *C. spixii*, but the latter two could be distinguished by their sequences.

## Genetic distances and barcode gap

Using the K2P model for COI Barcode, the intraspecific distances for each species varied from 0.000 to 0.014, with *N. grandicassis* having the highest value. The interspecific distances varied from 0.000 to 0.187, with the lowest values observed between *N. phrygiatus* and *N. quadriscutis* (0.000) and *C. agassizii* and *C. spixii* (0.010) (Table S4).

For Cytb, the intraspecific distances for each species ranged from 0.000 to 0.012, with *B. bagre* showing the highest value. The interspecific distances ranged from 0.000 to 0.203, with the lowest values observed between *N. phrygiatus* and *N. quadriscutis* (0.000) and *C. agassizii* and *C. spixii* (0.016) (Table S4).

Regarding the Barcode gap, the species of *Sciades*, *B. bagre*, *N. grandicassis* and *N. rugispinis* had enough genetic distances to distinguish them from the other species. For the other species, *N. phrygiatus*, *N. quadriscutis*, *C. agassizii* and *C. spixii*, there was no Barcode gap (Fig. 2).

## Species delimitation

The phylogenetic tree, based on Bayesian inference, was effective in discriminating the majority of different species, represented by reciprocally monophyletic groups, supported by high statistical values, both for COI and Cytb. However, the delimitation tests (GMYC, PTP and ASAP) presented different results for each marker, highlighting Cytb as the marker with the highest resolution to identify marine catfish. From the COI ultrametric tree, 10 MOTUS were retrieved, with all tests in agreement (Fig. 3A). For the Cytb tree, the GMYC and PTP tests delimited 11 MOTUS and the ASAP 10 (Fig. 3B).

The *Cathorops* genus showed different formation according to the marker and test (Fig. 3). The COI marker (Fig. 3A) generated a single MOTU for the species *C. agassizii*

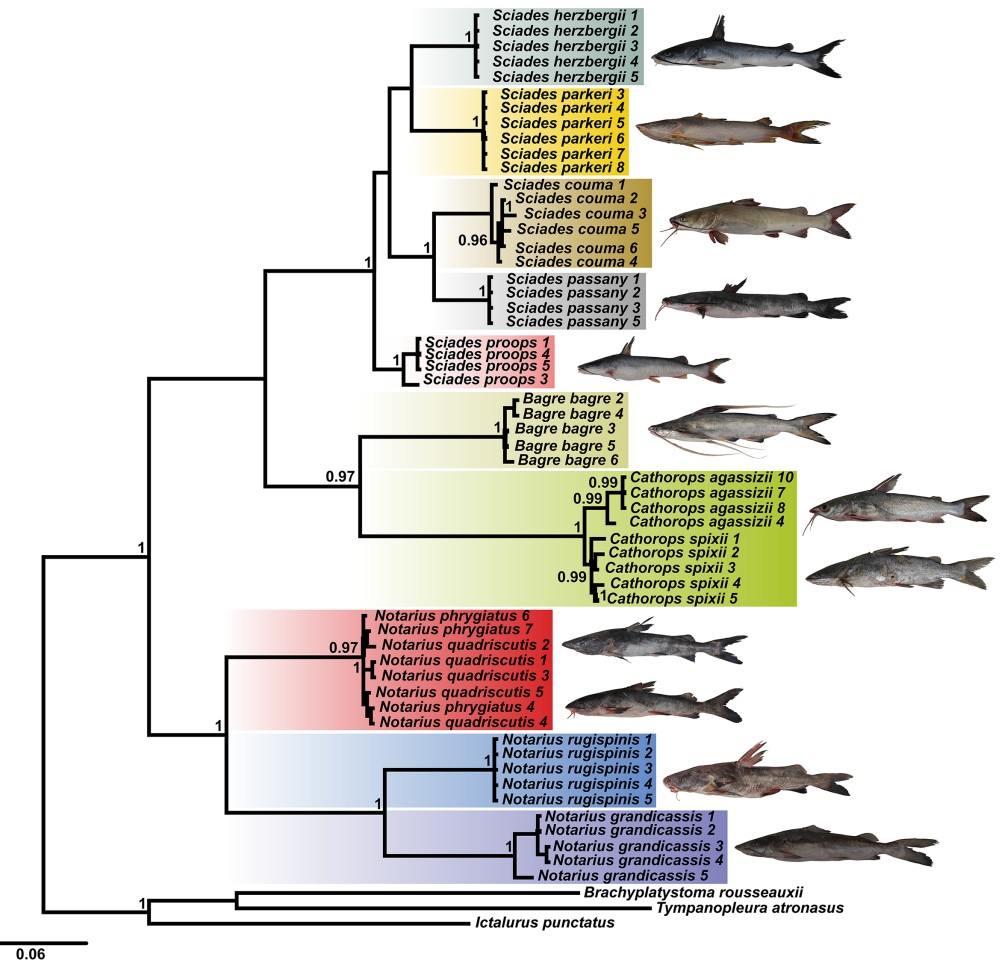

**Figure 1 Bayesian inference tree (BI) based on the diversity of Ariidae from the coastal region of the Amazon based on the concatenated mitochondrial genes COI and Cytb.** Bootstrap values are presented at the tree's internodes.      

and *C. spixii* for the three tests used. On the other hand, in the Cytb marker (Fig. 3B), the *C. agassizii* and *C. spixii* species formed different MOTUS in the GMYC and PTP tests, but recovered a single MOTU in the ASAP test (Fig. 4). The species *N. quadriscutis* and *N. phrygiatus* were not distinguished as separate taxa, considering both markers.

# DISCUSSION

## Diversity of Ariidae in the coastal Amazon

In the Amazon coastal region, 13 species of the Ariidae family are known to occur, eight of these species are endemic to the Amazon and Orinoco rivers plumes, and 12 (92%) of them were studied in this work (*Marceniuk et al., 2021*; *Marceniuk, Oliveira & Ferraris, 2023*) and are representative of the diversity of marine catfish on the North Brazilian coast. These species have widely shared morphological characters (*Marceniuk et al., 2021*), and therefore can be easily confused, in addition to the profound genetic similarity between some taxa, which can limit correct discrimination for some species. In an attempt to

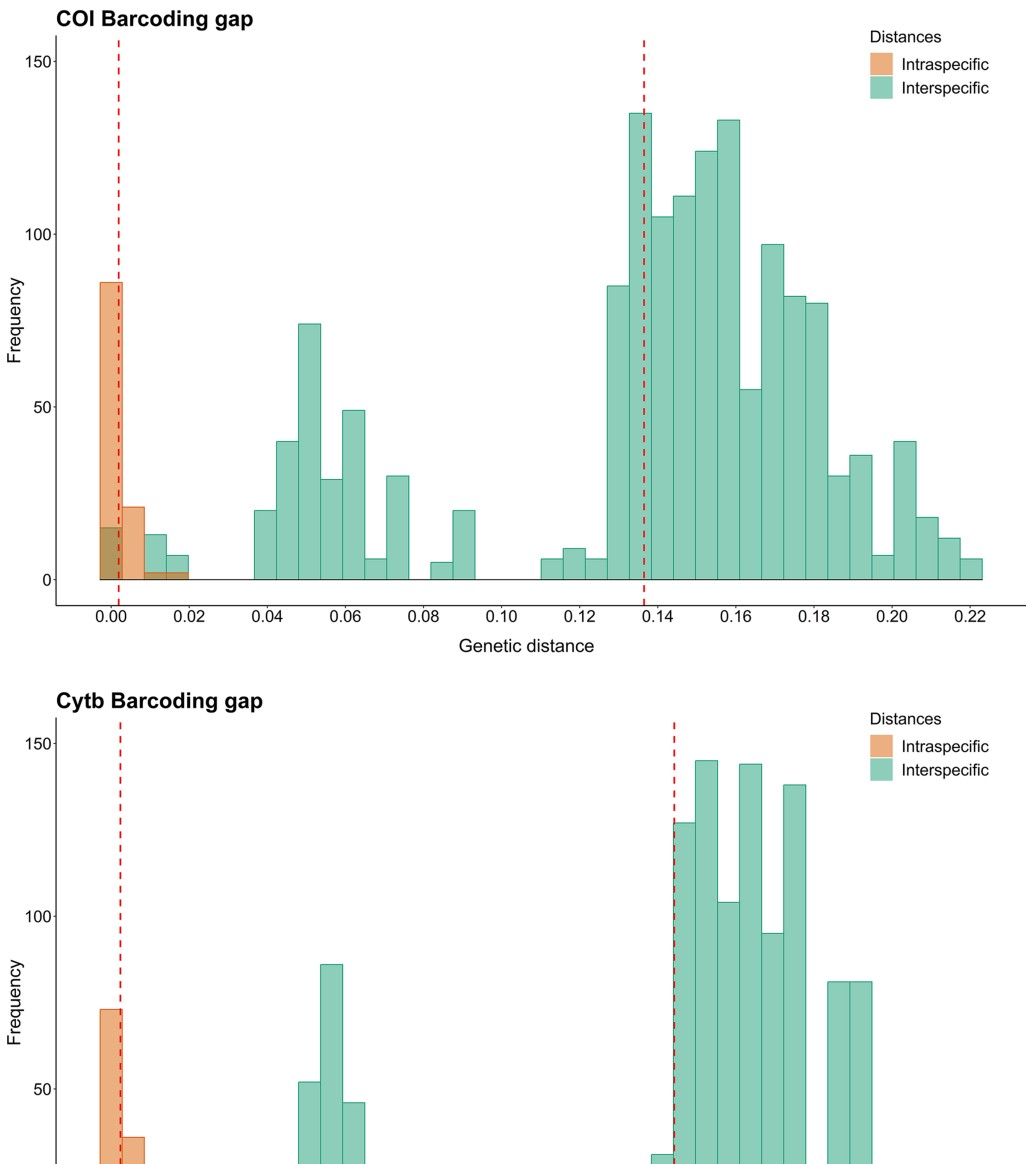

**Figure 2 Barcode gap histograms from COI and Cytb markers for Ariidae species from the coastal region of the Amazon.** Dashed red lines represent the averages of intra and interspecific genetic distances.

standardize a DNA Barcode marker for Amazonian coastal catfish, we corroborate data already available in the literature, which suggest a still confusing systematics for Ariidae, including many taxa from the South Western Atlantic.

Recent work has demonstrated that molecular identification approaches, using a single mitochondrial locus, for families of fish with recent diversification and poorly understood

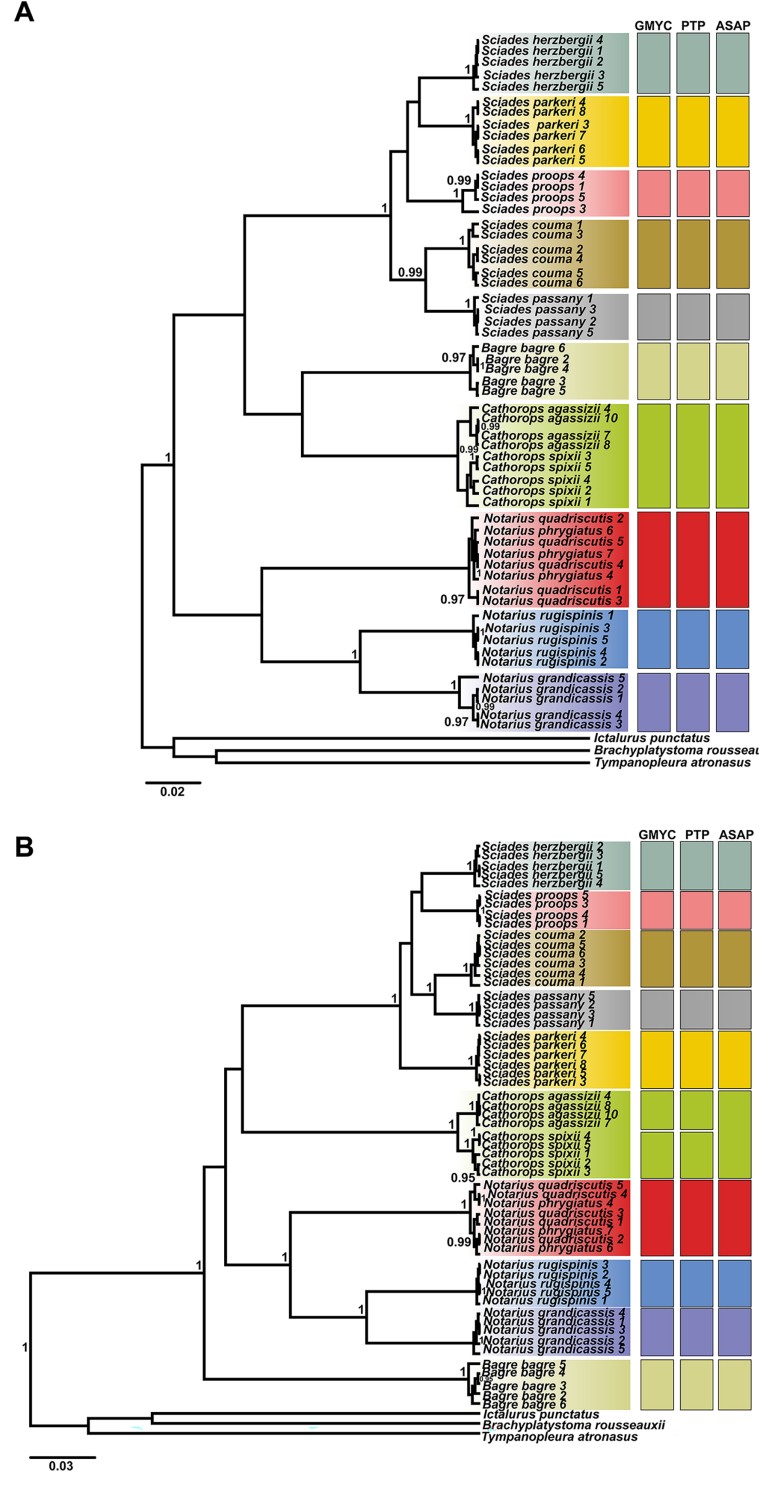

**Figure 3 Ultrametric topologies with species delimitation scenarios based on the mitochondrial genes COI (A) and Cytb (B), for Ariidae species from the coastal region of the Amazon.** Vertical bars correspond to each test. Different colors indicate each species discriminated. Bootstrap values are presented at the tree's internodes.

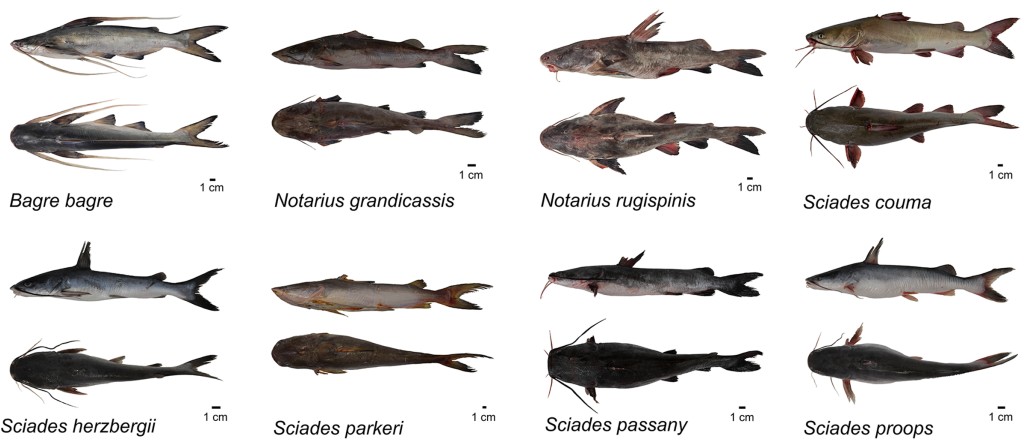

**Figure 4 Species of Ariidae from the coastal region of the Amazon that presented molecular identification according to morphological identification.**

taxonomy, may not be effective for some taxa, as already recorded for certain groups of Teleosts and Elasmobranchs (*Sanches et al., 2021*; *Santana et al., 2023*).

## Morphological identification *versus* molecular identification

High morphological similarity, ontogenetic variation and sexual dimorphism are effective barriers to the recognition of marine catfish species from the Ariidae family (*Marceniuk, 2005*, *2007*; *Marceniuk & Menezes, 2007*; *Betancur-R, Marceniuk & Béarez, 2008*). This scenario generated provisional generic classifications carried out in recent decades when they were based entirely on external morphological characters (*Taylor & Menezes, 1978*; *Kailola & Bussing, 1995*). Since then, studies that addressed genus changes, reevaluation and validation of species have been produced, showing the difficulties in understanding the true systematics of Ariidae (*Betancur-R et al., 2007*; *Marceniuk & Menezes, 2007*; *Betancur-R, Marceniuk & Béarez, 2008*; *Marceniuk, Menezes & Britto, 2012a*; *Marceniuk et al., 2012b*, *2017b*).

In this work, based on morphology, our previous identification recorded 12 species of marine/coastal catfish from the 57 Ariidae individuals sampled from the Amazon coastal region. The only species from the coastal Amazon not represented in this sampling was *C. arenatus*, a "Uricica amarela" very similar morphologically and genetically to *C. spixii*, however, with distribution restricted of the Amazon and Orinoco rivers plumes (*Marceniuk et al., 2021*; *Marceniuk, Oliveira & Ferraris, 2023*).

The phylogenetic tree based on Bayesian inference, using combined data (COI and Cytb) revealed 10 main clades (Fig. 1). The groups composed of the species of the genera *Sciades* and the species *B. bagre*, *N. grandicassis*, *N. rugispinis*, shown in Fig. 4, had molecular identification that matched the morphological identification and their relationships agreed with *Betancur-R et al. (2007)* and *Marceniuk, Menezes & Britto (2012a)*. Identification also confirmed by the similarities obtained in the comparisons of public databases (Tables S2 and S3).

The phylogenetic analysis revealed a discrepancy between the morphological and molecular identification of *N. quadriscutis* and *N. phrygiatus*, two species that clustered together in the phylogenetic trees. The genetic analysis showed no Barcode gap between *N. quadriscutis* and *N. phrygiatus*, with zero genetic distance for both COI and Cytb. This result was consistent with the delimitation tests, where the two species formed a single clade, and also with the fact that they shared mitochondrial haplotypes (Fig. 4, Table S1).

*N. quadriscutis* and *N. phrygiatus* are two distinct species of catfish that can be easily identified by their external morphology. *N. quadriscutis* has a broad, square-shaped nuchal plate that resembles a "butterfly", while *N. phrygiatus* has a narrow, V-shaped nuchal plate and lacks the vomerine tooth plates (*Marceniuk et al., 2017a*). We compiled images of these two species, showing their different characteristics (Fig. 5A) and the polymorphic sites at each marker (Fig. 5B).

However, in the Bayesian inference analysis, public COI samples for *N. phrygiatus* did not group with our sequences, forming a separate clade (Fig. 5C). In fact, the molecular analysis by *Marceniuk, Oliveira & Ferraris (2023)* using mitochondrial and nuclear markers confirmed that *N. quadriscutis* and *N. phrygiatus* belong to separate clades, validating their taxonomic status.

This result raises some hypotheses. First, no matter how closely we followed the literature, our morphological identification of *N. phrygiatus* was erroneous. Second, public deposits of *N. phrygiatus* are misidentified. Finally, there is possibly a process of hybridization or mitochondrial introgression between the two species, *N. quadriscutis* and *N. phrygiatus*, reinforced by the species' distribution zones (Fig. 5D), with an area of overlap and then sympatry. We believe that this last hypothesis, hybridization, is the most likely in this case. In addition, only a few sequences were available for Cytb and none for COI. More sequences were needed for a better understanding.

The relationships between the genera *Amphiarius*, *Aspistor* and *Notarius* are confusing. *Amphiarius* species differ from *Aspistor* and almost all *Notarius* species in swim bladder morphology, but this character is shared between *Amphiarius* and *N. grandicassis* (*Marceniuk & Birindelli, 2010*). Therefore, the monophyly of the genus *Amphiarius* was suggested by *Marceniuk (2003)* and *Kailola (2004)*, representing the description of a new genus and thus forming three valid genera, *Amphiarius*, *Aspistor* and *Notarius*, according to morphological characters (*Marceniuk & Menezes, 2007*).

The species sampled in the present study, *N. phrygiatus* and *N. rugispinis* belonged to the genus *Amphiarius*, while *N. quadriscutis* belonged to the genus *Aspistor*. However, *Betancur-R et al. (2007)* considered that the genera *Amphiarius* and *Aspistor* are synonymous with *Notarius*, the latter being a valid genus, a classification also already adopted in the work of *Marceniuk, Oliveira & Ferraris (2023)*, where a new classification for Ariidae was proposed.

The haplotypes of *N. phrygiatus* and *N. quadriscutis* showed 99% similarity with *N. luniscutis* in GenBank (Table S2). Although *N. luniscutis* is not found on the northern coast of Brazil, but in the Northeast, East and South regions (*Marceniuk et al., 2019*), *N. quadriscutis* and *N. luniscutis* represent distinct morphological lineages, but with an absence of genetic differentiation in the Cytb genes and subunits 8 and 6 of ATP synthase

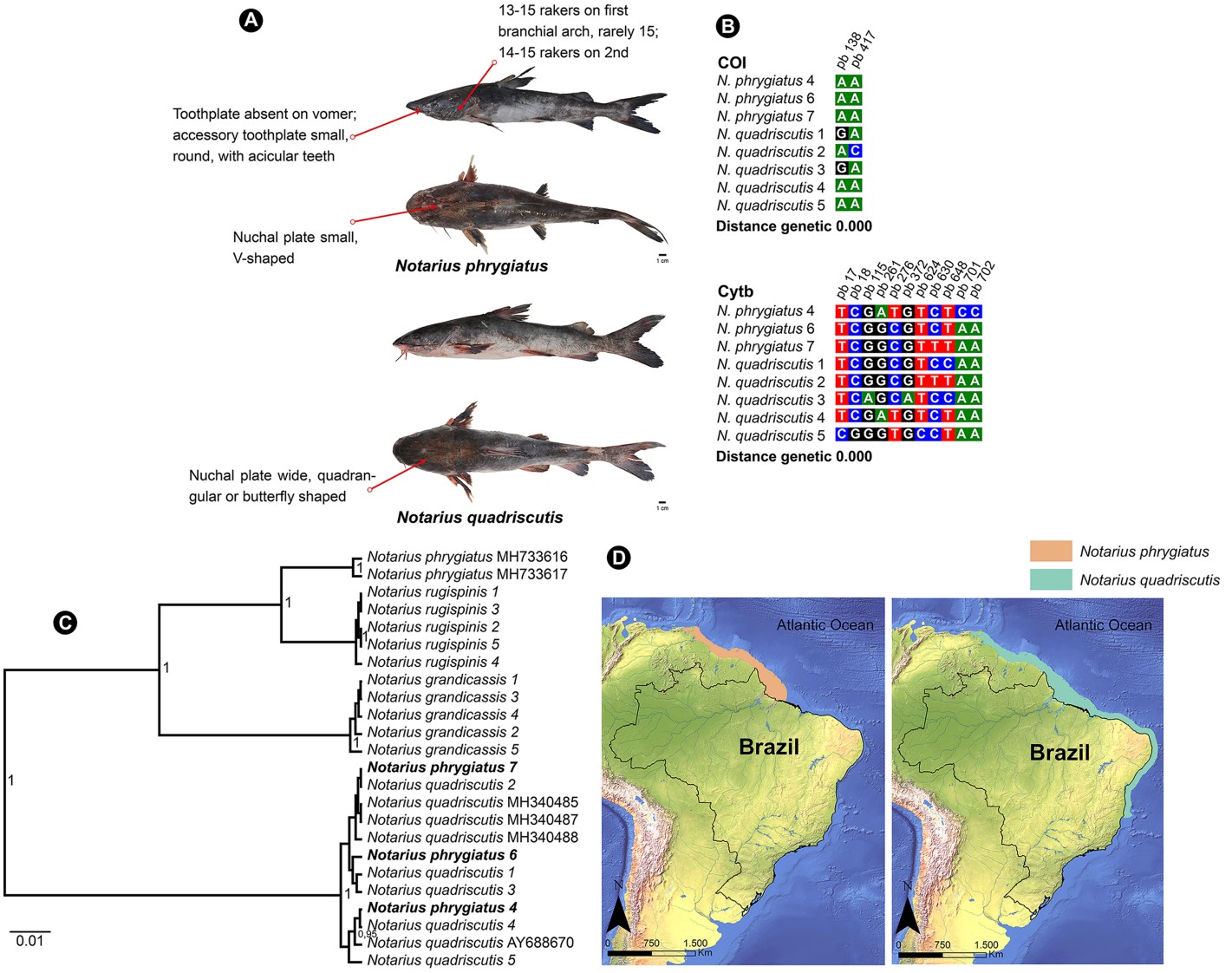

**Figure 5 Morphological and molecular comparison and distribution area between Ariidae species from the coastal region of the Amazon,** ***Notarius phrygiatus* and *Notarius quadriscutis*.** (A) Main morphological characters that diûerentiate the two species according to *Marceniuk et al. (2017a)*. (B) Polymorphic sites between the two species for the mitochondrial genes COI and Cytb. The numbering above the nucleotides reûects the position of the polymorphic sites in the 505 bp fragments for COI and 704 bp fragments for Cytb. (C) Bayesian Inference Tree (BI) of *Notarius* species. (D) Accepted distribution range according to International Union for Conservation of Nature (IUCN) for both species. The base shape file used was obtained from the Natural Earth website (https://www.naturalearthdata.com). Species distribution shape files were obtained from IUCN (https://www.iucnredlist.org/species/197078/2478828/https://www.iucnredlist.org/species/190119/1940944).

(ATPase 8/6), that is, the two species undergo a very recent speciation process (*Marceniuk et al., 2019*), which is also found in other Ariidae (*Betancur-R & Armbruster, 2009*).

As *N. phrygiatus* and *N. quadriscutis* formed a single clade (Figs. 2–4), 99% similarity of the two species with *N. luniscutis* was expected.

However, the 99% similarity of *N. phrygiatus* and *N. quadriscutis* with *N. grandicassis* on the BOLD platform (Table S3) is a consequence of incorrect identification, considering

that *N. phrygiatus* and *N. quadriscutis* did not group with *N. grandicassis* in our analyzes (Figs. 2–4), indicating that they are distinct species.

## Species delimitation

The genus *Cathorops* presented different species delimitation according to the marker and the test (Fig. 3). The COI marker generated a single MOTU for the species *C. agassizii* and *C. spixii* in the three tests used. On the other hand, in the Cytb marker, *C. agassizii* and *C. spixii* formed a single MOTU in the ASAP test and two MOTUs in the GMYC and PTP tests. Unlike the GMYC and PTP coalescence-based tests, the clades formed from the ASAP test are generated from a pairwise genetic distance matrix (*Puillandre, Brouillet & Achaz, 2021*), which resulted in the formation of a single clade for both species analyzed.

   *Cathorops* is a monophyletic group according to morphological and molecular data (*Betancur-R, 2009*; *Marceniuk, Menezes & Britto, 2012a*). *C. agassizii* and *C. spixii* present morphological characteristics that separate the two species (*Marceniuk et al., 2012b*) (Fig. 6), however, they present an interspecific distance of 1% with the COI, a value lower than the 2% threshold used to differentiate species (*Hebert & Gregory, 2005*) and 1.6% with Cytb. Interspecific distances below 2% for the Cytb gene, in species of the genus *Cathorops*, have already been observed in *Cathorops fuerthii* and *Cathorops maglarensis*, which have 1.3% genetic distance in the Cytb gene, and *Cathorops mapale* and *Cathorops wayuu*, which have 1.1% distance between them in ATPase 8/6 gene (*Marceniuk et al., 2019*).

   Our results showed a low interspecific divergence in the pair of species of the genus *Cathorops*, but without the sharing of haplotypes for the two markers, which has already been observed in other studies (*Betancur-R et al., 2007*), which may also be indicative of recent speciation (*Cawthorn, Steinman & Corli Witthuhn, 2011*; *Hou et al., 2018*), or that it is a single species of "Uricica". This low genetic distance between species of the genus *Cathorops* results in the similarity of more than 99% of *C. agassizii* haplotypes to the species *C. spixii* and *C. arenatus* (Tables S1 and S2). Further investigations using nuclear marker libraries should be conducted to elucidate relationships within the genus *Cathorops*, especially involving *C. agassizii*, *C. spixii* and *C. arenatus*.

## COI and Cytb as barcodes for Ariidae

The sampled individuals of *N. grandicassis* formed a single clade in the concatenated tree (Fig. 1) and in the delimitation tests (Fig. 3), but they had the highest intraspecific distance (1.4%) among all the individuals we analyzed in this study (Table S3). This distance was lower than the one reported by *Guimarães-Costa et al. (2020)*, who found a distance of 3.2% for *N. grandicassis* individuals, resulting in two operational taxonomic units (OTUs). These results indicate the potential presence of cryptic diversity in the Amazon coastal region (*Guimarães-Costa et al., 2020*), which needs further evaluation along with the other species in the genus *Notarius*, considering the low genetic diversity between *N. grandicassis* and *Notarius parmocassis*, the latter being common on the East coast of Brazil (*Marceniuk et al., 2017b*).

   In our analyses, an overlap of intra and interspecific distances was seen, resulting in the absence of a gap between species, a scenario caused mainly by the clade formed by

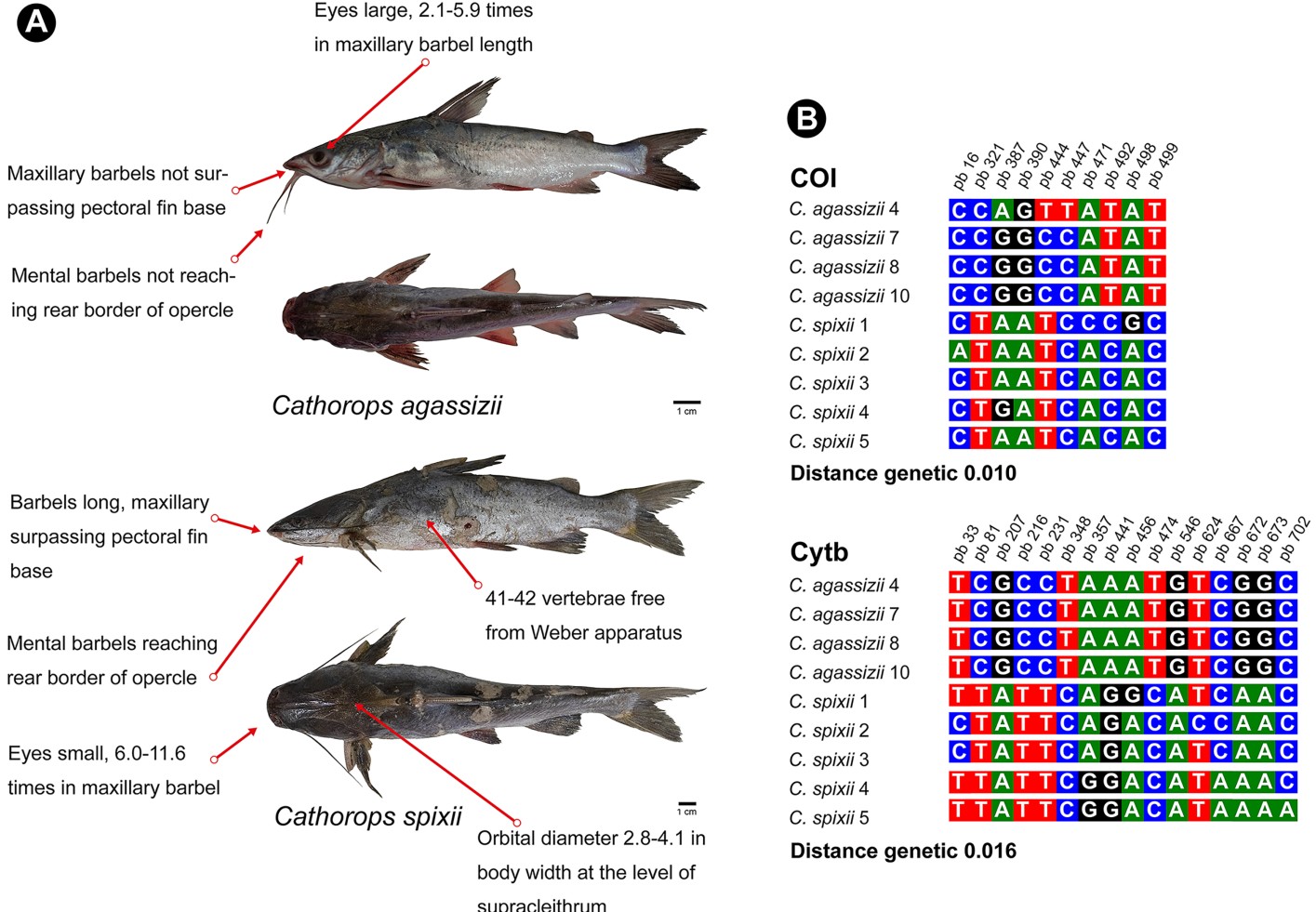

**Figure 6 Morphological and molecular comparison between the Ariidae species from the coastal region of the Amazon, *Cathorops agassizii* and *Cathorops spixii*.** (A) Main morphological characters that differentiate the two species according to *Marceniuk et al. (2017a)*. (B) Polymorphic sites between the two species for the mitochondrial genes COI and Cytb. The numbering above the nucleotides reflects the position of the polymorphic sites in the 505 bp fragments for COI and 704 bp fragments for Cytb.

*N. quadriscutis* and *N. phrygiatus* (Fig. 1). Therefore, the COI and Cytb markers are not effective to be DNA Barcode for all Ariidae species, despite well-resolved clades, mainly the genera *Bagre* and *Sciades* (Fig. 1).

The COI and Cytb genes are commonly used to investigate fish diversity, functioning as a DNA Barcode tool (*Fernandes et al., 2017*; *Bingpeng et al., 2018*; *Wang et al., 2018*; *Gomes et al., 2019*), and the use of these two markers is linked to the concept of Barcode gap, that is, the greatest intraspecific distance should not exceed the smallest interspecific distance, separating the species (*Meyer & Paulay, 2005*; *Knebelsberger et al., 2014*; *Shen et al., 2016*).

Unlike COI, the Cytb gene presented itself as a better Barcode for Ariidae fish from the coastal region of the Amazon, presenting higher interspecific distance values (Table S3) and forming clades different from *Cathorops* species in the GMYC and PTP tests (Fig. 3B). The Cytb gene also presented more polymorphic sites between the two species, *C. agassizii* and *C. spixii* (Fig. 6B). The Cytb gene has been used as a barcode for different groups of

fish, successfully identifying species of Flatfishes, Gadoids, Scombridae, Stromateidae, among others (*Sotelo et al., 2001*; *Calo-Mata et al., 2003*; *Chow et al., 2003*; *Wei et al., 2021*).

To improve the identification of species in complex groups like the genus *Cathorops*, it may be necessary to adopt Cytb or other new markers instead of the COI gene, which is the most widely used Barcode fragment. The COI gene may not provide enough information to resolve the systematics of these groups. Nuclear markers should also be included to avoid the limitations of ancestral polymorphism in the analyses.

## CONCLUSIONS

We conducted a study to examine the diversity of Ariidae in the Amazon coastal area using morphological and molecular methods (COI and Cytb). Most taxa, such as *Sciades* and *Bagre* genera, were consistent with their previous morphological identification based on the molecular data. However, *C. agassizii* and *C. spixii*, which were also supported by morphology, had low genetic variation and need more data to confirm their status. *N. grandicassis* had the highest intraspecific distance values, suggesting cryptic diversity within this species. Despite their morphological differences, *N. quadriscutis* and *N. phrygiatus* formed a single clade, indicating case of mitochondrial introgression, reinforced by the species' distribution areas.

Our results provided new insights for a group with poorly understood systematics such as marine catfishes from the Ariidae family which limited the effectiveness of mitochondrial markers to distinguish between species. However, for most of the taxa we analyzed, COI and Cytb were efficient as DNA Barcode, especially Cytb, which had more polymorphism and resolution. The molecular tools we used were very effective for discriminating and identifying most species. The unresolved cases are related to the unresolved systematics of the group and require a standardized nuclear marker to identify the species.

### Funding

This work was supported by the Conselho Nacional de Desenvolvimento Científico e Tecnológico (CNPq), under the Universal Call (process: 439113/2018-0) and from Coordenação de Aperfeiçoamento de Pessoal de Nível Superior (CAPES), which provided a doctoral scholarship (88887.494974/2020-00) for Ítalo Lutz. The APC was covered by the Programa de Apoio à Publicação Qualificada (PAPQ)/Universidade Federal do Pará (UFPA). The funders had no role in study design, data collection and analysis, decision to publish, or preparation of the manuscript.

### Grant Disclosures

The following grant information was disclosed by the authors:
Conselho Nacional de Desenvolvimento Científico e Tecnológico (CNPq): 439113/2018-0.

Coordenação de Aperfeiçoamento de Pessoal de Nível Superior (CAPES): 88887.494974/2020-00.

Programa de Apoio à Publicação Qualificada (PAPQ)/Universidade Federal do Pará (UFPA).

## Competing Interests

The authors declare that they have no competing interests.

## Author Contributions

- Ítalo Lutz conceived and designed the experiments, performed the experiments, analyzed the data, prepared figures and/or tables, authored or reviewed drafts of the article, and approved the final draft.
- Thais Martins analyzed the data, prepared figures and/or tables, authored or reviewed drafts of the article, and approved the final draft.
- Paula Santana analyzed the data, authored or reviewed drafts of the article, and approved the final draft.
- Charles Ferreira analyzed the data, prepared figures and/or tables, and approved the final draft.
- Josy Miranda performed the experiments, prepared figures and/or tables, and approved the final draft.
- Suane Matos performed the experiments, prepared figures and/or tables, and approved the final draft.
- Valdemiro Muhala analyzed the data, authored or reviewed drafts of the article, and approved the final draft.
- Iracilda Sampaio conceived and designed the experiments, authored or reviewed drafts of the article, and approved the final draft.
- Marcelo Vallinoto conceived and designed the experiments, authored or reviewed drafts of the article, and approved the final draft.
- Grazielle Evangelista-Gomes conceived and designed the experiments, analyzed the data, authored or reviewed drafts of the article, and approved the final draft.

## Animal Ethics

The following information was supplied relating to ethical approvals (*i.e.*, approving body and any reference numbers):

All samples were collected and transported with authorization of the Instituto Chico Mendes de Conservação da Biodiversidade (ICMBio) (SISBIO license n. 12773-1).

## Data Availability

Sequences are available at GenBank, specifically: The COI sequences: OR646749 to OR646805, and the Cytb sequences: OR674791 to OR674847.

The COI sequences are also available at the Barcode of Life Data System (BOLD): BAGR001-24 to BAGR057-24.

## Supplemental Information

Supplemental information for this article can be found online at http://dx.doi.org/10.7717/peerj.17581#supplemental-information.

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
