# Peer review of "Marine catfishes (Ariidae—Siluriformes) from the Coastal Amazon: mitochondrial DNA barcode for a recent diversification group?"

_PeerJ, doi:10.7717/peerj.17581_

## Round 0.1 · original submission · Major Revisions

After careful evaluation of your revised manuscript, consider the comments of Reviewers 1 and 2.

I agree with reviewer 2. The methodology should be more detailed regarding aspects of species morphology. Review the manuscript and answer the questions raised by both reviewers, especially regarding the validity of the findings and additional comments. Then, resubmit the manuscript within 35 days.

**Language Note:** The review process has identified that the English language must be improved. PeerJ can provide language editing services - please contact us at [email protected] for pricing (be sure to provide your manuscript number and title). Alternatively, you should make your own arrangements to improve the language quality and provide details in your response letter. – PeerJ Staff

·

Basic reporting

This paper describes a study on the Marine catfshes (Ariidae 3 Siluriformes) from the Coastal Amazon: Mitochondrial DNA barcode for a recent divarication group. The data is a useful contribution to the mitogenome. This provides taxonomic and phylogenetic studies of Marine catfishes fish.

Experimental design

Author analyzed the two mitochondrial gene COI and Cytb. This is good for the species identification and phylogenetic relationship.

Validity of the findings

Author has done many calculations. Phylogenetic analysis, DNA barcode and genetic distance was done. Analysis is good. They are saying that there is no genetic variation between N. quadriscutis and N. phrygiatus. This is need to be more confirmation with more samples from different locations morphology and molecular. If the sequences is present in NCBI. You can compare your data with NCBI data. I think author improved the paper and it should be published.

Additional comments

The quality of the English presentation does not meet the journal's requirements. There are many typos, grammar errors, and awkward sentences. Help from a professional language editing service or a fluent English speaker is needed.

Like : access- it should be accession number

Reviewer 2 ·

Basic reporting

The manuscript entitled "Marine catfishes (Ariidae ñ Siluriformes) from the Coastal Amazon: Mitochondrial DNA barcode for a recent diversification group?" The study conducted an important methodology, which is highly relevant and necessary. However, a crucial point to be revised is regarding the morphological identification methodology. The authors do not explain how the identifications were made. Since this is one of the study's objectives, I request that the methodological description be included in the manuscript. I emphasize the importance of including this information, as it is one of the two main objectives of the study. The molecular methodology was very well described; however, the morphological identification methodology was not even mentioned. Because of this, I accept the manuscript with major revisions.

Experimental design

Please specify in details how the morphological analyses were conducted.

Validity of the findings

The study results are very interesting and scientifically relevant.
Have the sequences resulting from the study been deposited in the BOLD database? If yes, please include the identifications. If not, I suggest that the authors deposit the sequences to assist in future studies on molecular identification of Amazonian fishes.

---

## Round 0.2 · accepted · Accept

Dear author,
According to the last review round, all the corrections and observations from the reviews were promptly added to the manuscript, and my final decision is to accept publishing in the PeerJ. Thank you for sending your research to us.
Best regards.

·

Basic reporting

The article has been improved. It now conforms to professional standards of courtesy and expression. The author has revised the paper thoroughly and addressed all the reviewed comments carefully. I am accepting this manuscript in its current form.

Experimental design

The research question is well-defined, relevant, and meaningful. The reviewed submission effectively addresses the provided comments and questions, which are also relevant and meaningful.

Validity of the findings

Only a few sequences are present in the global database for the studied species. Therefore, the generated data in this study will be valuable for researchers worldwide. The conclusions are appropriately stated and connected to the original question investigated. Hence, it should be published in its present form

Reviewer 2 ·

Basic reporting

The manuscript entitled "Marine catfishes (Ariidae - Siluriformes) from the Coastal Amazon: Mitochondrial DNA barcode for a recent diversification group?" The study conducted an important methodology, which is highly relevant and necessary. All requested changes were carefully made, and after the review, I encourage the acceptance of the manuscript for publication.

Experimental design

All requested changes have been made.

Validity of the findings

The study conducted an important methodology, which is highly relevant and necessary.